# Impact and prognosis of the expression of IFN-α among tuberculosis patients

Vibha Taneja[1,2,3], Priya Kalra[3], Manish Goel[4], Gopi Chand Khilnani[4¤], Vikram Saini[3], G. B. K. S. Prasad[2], Umesh Datta Gupta[1], Hanumanthappa Krishna Prasad[3]*

**1** National JALMA Institute of Leprosy and Other Mycobacterial Diseases, Tajganj, Agra, India, **2** Department of Biochemistry, Jiwaji University, Gwalior, Madhya Pradesh, India, **3** Department of Biotechnology, All India Institute of Medical Sciences, New Delhi, India, **4** Department of Pulmonary, Critical Care and Sleep Medicine, All India Institute of Medical Sciences, New Delhi, India

¤ Current address: Chairman, PSRI Institute of Pulmonary, Critical Care and Sleep Medicine, PSRI Hospital, New Delhi, India
* hk_prasad@hotmail.com, hkp1000@gmail.com

**Data Availability Statement:** All relevant data are within the manuscript and its Supporting Information files.

**Funding:** The fellowship to VT from Indian Council of Medical Research (ICMR), Government of India;

## Abstract

*Mycobacterium tuberculosis* (*M.tb*) infection stimulates the release of cytokines, including interferons (IFNs). IFNs are initiators, regulators, and effectors of innate and adaptive immunity. Accordingly, the expression levels of Type I (α, β) and II (γ) IFNs, among untreated tuberculosis (TB) patients and household contacts (HHC) clinically free of TB was assessed. A total of 264 individuals (TB patients-123; HHC-86; laboratory volunteers-55; Treated TB patients-36) were enrolled for this study. IFN-α mRNA expression levels predominated compared to IFN-γ and IFN-β among untreated TB patients. IFN-α transcripts were ~3.5 folds higher in TB patients compared to HHC, ($p$<0.0001). High expression of IFN-α was seen among 46% (56/ 123) of the TB patients and 26%, (22/86) of HHCs. The expression levels of IFN-α correlated with that of IFN transcriptional release factor 7 (IRF) ($p$<0.0001). In contrast, an inverse relationship exists between PGE2 and IFN-α expression levels; high IFN-α expressers were associated with low levels of PGE2 and vice-versa (Spearman's rho = -0.563; $p$<0.0001). *In-vitro*, IFN-α failed to restrict the replication of intracellular *M.tb*. The anti-mycobacterial activity of IFN-γ was compromised in the presence of IFN-α, but not by IFN-β. The expression of IFN-α and β diminished or is absent, among successfully treated TB patients. These observations suggest the utility of assessment of Type I IFNs expression levels as a prognostic marker to monitor tuberculosis patient response to chemotherapy because changes in Type I IFNs expression are expected to precede the clearance and /reduction in bacterial load.

## Introduction

Tuberculosis (TB) remains a global public health problem. Annually TB accounts for $>10 \times 10^6$ new cases, and ~1.2 x $10^6$ deaths [1]. A large proportion of individuals exposed to *M. tuberculosis* (*M.tb*) generate immune responses that clear the infecting bacilli or drive them to dormancy resulting in latent TB infection (LTBI). Approximately 10% of these individuals with LTBI later reactivate to develop clinical disease. Protective immunity appears to be associated

HKP Emeritus Medical Scientist, ICMR,; Financial support of the ICMR grant "Study of Immunological impact of type I Interferon (s) on immune response in tuberculosis patients," project ID : 2013-0063.

**Competing interests:** The authors have declared that no competing interests exist.

with the generation of antigen-specific CD4$^+$ T cells expressing IFN-γ (Type II IFN), including IL-12 and TNF-α. In contrast to the well established protective role of IFN-γ, type I IFNs can be either protective or detrimental in bacterial infections [2]. Type I IFNs are primarily involved in the immune response to viral infections. IFN-α comprises of multiple forms, whereas, IFN-β is a single type. Type I IFNs can potentially exacerbate pathogenesis in chronic viral infections via immunosuppression or inflammation and tissue destruction [3]. On the one hand, Type I IFNs (IFN-α/β) are potent inhibitors of IL-12 production by human monocytes/macrophages, a critical cytokine required for the induction of IFN-γ [4]; on the other hand, they can induce IFN-γ production by T and NK cells in an IL-12 independent manner [5]. IFNα/β reduces monocyte viability, compromises their bacteriostatic activity, and antigen presentation ability [6]. However, IFN-β enhances BCG immunogenicity by facilitating dendritic cell (DC) cell maturation [7]. Type I IFNs have been administered as an adjunctive therapeutic agent to PTB patients harboring multi-drug resistant *M.tb* strains [8,9]. Several studies have reported that the induction of Type I IFNs precedes the onset of clinical tuberculosis [10–13]. Thus the design of the current study included: 1) assessment of the levels of Type I (IFN-α and β) and II (IFN-γ) IFNs among Indian TB patients, and 2) to examine the effect of Type I IFN in modulating replication of *M.tb* resident in human macrophages.

## Materials and methods

### Study subjects

TB patients were enrolled from the Out-patient Department of Pulmonary, Critical Care and Sleep Medicine, AIIMS, New Delhi. Of the 123 TB patients recruited, 56 were pulmonary (PTB) and 67 were extra-pulmonary TB (EPTB) patients. Healthy family contacts (HHC, n = 86) of the patients were also recruited for participation in the study. Thirty six/123 TB patients were available for follow-up after successful treatment; healthy laboratory volunteers (HV, n = 55) were recruited from the Department of Biotechnology, AIIMS, New Delhi were included as controls. Informed consent was taken from all individuals, and the study has been approved by the Ethics Committee of AIIMS (IEC/NP-196/2013, OP-01/10.04.2015).

### RNA isolation and cDNA synthesis

Sera and RNA were obtained from peripheral blood samples. For the latter, RBCs were lysed on ice using ACK lysis buffer (154.4 mM ammonium chloride, 10 mM potassium bicarbonate, 97.3 μM EDTA tetrasodium salt for 1L) for 20 minutes and the leukocyte pellet suspended in TRIzol reagent was processed for RNA extraction, (Promega Co., Wisconsin, MD, USA). The extracted RNA was treated with DNase I (Promega, Wisconsin, MD, USA), cleaned by using the RNeasy Blood Mini Kit (Qiagen, Hilden, Germany) according to the manufacturer's instructions. DNA-free RNA (1μg) from each sample was reverse transcribed using the Maxima First Strand cDNA Synthesis kit (ThermoFischer Scientific, Rockford, USA), according to the manufacturer's instructions. For serum, clotted blood samples were processed as per the routine procedure. The separated sera were aliquoted and stored at -20˚C.

### Real-time PCR

The cDNA obtained was subjected to Real-Time PCR analysis of IFN-α, IFN-β, IFN-γ, IRF5, and IRF7 mRNA expression using primers detailed in Table 1.

PCR master reaction mix (25μl) was setup containing Maxima® SYBR Green / ROX qPCR Master Mix (ThermoFischer Scientific, Rockford, USA), 0.5 μM of each primer, and 100ng of cDNA / sample. Real-time detection of transcripts was carried out in MyIQ cycler (Bio-Rad,

**Table 1. Primers used in the study.**

| Target | Primer | | Reference |
|---|---|---|---|
| | Forward | Reverse | |
| IFN-α | 5′- GCTGAATGACCTGGAAGCCTGTG -3′ | 5′- GGGAGGTTGTCAGAGCAGAAATC-3′ | Verma et al., 2012 [14] |
| IFN-β | 5′-AAGGAGGACGCCG CATTGAC-3′ | 5′-ATAGACATTAGCCAGGAGGTTC-3′ | Self-designed |
| IFN-γ | 5′- TCGTTTTGGGTTCTCTTGGC-3′ | 5′-TCCGCTACATCTGAATGACC-3′ | Self-designed |
| IRF5 | 5′-CAGGACGGAGATAACACCAT-3′ | 5′-GGTGTATTTCCCTGTCTCCT-3′ | Self-designed |
| IRF7 | 5′-AAAACCAACTTCCGCTGC-3′ | 5′-GCCTCAGTCTGGTCCGTGC-3′ | Self-designed |
| β-actin | 5′- CGGCATCGTCACCAACTGG-3′ | 5′- ACGTTGCTATCCAGGCTGTGC-3′ | Verma et al., 2012 [14] |
| IFI44 | 5′-GAGAGATGTGAGCCTGTGAGG-3′ | 5′-TTTTCCTTGTGCACAGTTGAT-3′ | Self-designed |
| FCγR1 | 5′-CTT CTC CTT CTA TGT GGG CAG T-3′ | 5′-GCT ACC TCG CAC CAG TAT GAT-3′ | Zhang et al., 2014 [15] |

California, U.S.A.) with cycling parameters as follows, denaturation at 94˚C for 5 min; 40 cycles each of denaturation at 94˚C for 15 sec, annealing at 60˚C for 30 sec and extension at 72˚C for 30 sec. The fluorescence signal was collected during the extension step. On completion of the run, the threshold cycle (Ct) values were determined. The melt curve was generated at a ramp rate of 2% (CFX Manager, v2.0, Bio-Rad, California, U.S.A.).

Assessment of IFN-α, IFN-β, IFN-γ, IRF5, and IRF7 mRNA expression in the samples derived from TB patients, their family contacts, and healthy volunteers was done. Accordingly, the normalized expression of IFN-α, IFN-β, IFN-γ, IRF5, and IRF7 was calculated from the threshold cycle (Ct) values normalized to β-actin Ct values. The normalized mRNA expression of IFN-α, IFN-β, IFN-γ, IRF5, and IRF7 relative to healthy volunteers was calculated as a $2^{-\Delta\Delta Ct}$ method [16]. The ratio between: IFN Transcripts determined in a patient / Group mean of IFN Transcripts for Healthy Volunteers; established the 'Fold' expression for the IFN in the individual. Accordingly in individuals wherein mRNA expression of an IFN was higher than the group mean of healthy volunteers (>1-fold) was considered as a high expression. The statistical significance of the fold mRNA expression between groups was determined using the Mann-Whitney test.

## ELISA

Serum Prostaglandin E2 levels were determined for 62 TB patients (24 PTB, 38 EPTB) and 18 family contacts with a commercially available kit (PGE2, Enzo Life Sciences Inc., NY, USA)

### *In vitro* infection of THP-1 cells with M.tb

Undifferentiated ($0.5 \times 10^6$) THP-1 (human monocytic) cells were seeded in 24 well plates, stimulated with 50ng/μl of Phorbol myristate acetate (PMA, Sigma Aldrich, Co., St Louis, USA) for 16 hours. The plate was washed and kept for 48 hours in $CO_2$ incubator for 48 hours. After a resting period of 48 hours, the PMA differentiated cells were exposed to live and heat-killed *M.tb* ($H_{37}R_V$) (*M.tb*: THP-1; MOI 10:1) for 4 hours. The extracellular bacteria were removed by washing with plain RPMI and the infected cells maintained with 10% RPMI for 0 ($T_0$), 2($T_2$), 4($T_4$), 6($T_6$), 8($T_8$), 16($T_{16}$), 24($T_{24}$), and 48 ($T_{48}$) hours respectively. RNA extraction, cDNA preparation, and RT-PCR for assessment of transcripts for IFN-α, IFN-β, and IFN-γ were performed at each time point, as described Above.

### Inhibition of mycobacterial growth assay

Differentiated THP-1 cells ($0.1 \times 10^6$) were seeded and infected, as mentioned above. In designated wells simultaneously 1μl of anti-IFN-α (Biorbyt, Cambridge, U.K.), anti-IFN-β and anti-

IFN-γ (Affymetrix, ThermoFischer Scientific, Rockford, U.S.A.) neutralizing antibodies were added individually or as a mixture of antibodies, (Treated cultures); untreated cultures served as Controls. The extracellular bacteria were removed 4 hrs post-infection, and cultures maintained with 10% RPMI alone in case of untreated control cultures; whereas in case of treated cultures respective antibody was added along with 10% RPMI for 5 days; after which cells were lysed and serial dilutions of lysates plated on 7H11 agar plates. Colony-forming units (CFUs) were obtained at five weeks post-plating. The neutralizing activity of the antibodies was confirmed by monitoring the inhibition of expression of the respective interferon-inducible genes namely IFI44 (IFN-α/β inducible gene [17,18] and FcγR1 (IFN-γ inducible gene [19,20], (S1 Fig). The primers used to monitor the expression of IFI44 and FcγR1 have been listed in Table 1.

## Results

### Enhanced IFN-α gene expression in TB patients compared to healthy household contacts (HHC)

The relative mRNA expression of IFNs α, β, and γ among TB patients and HHC is presented in Fig 1 and Table 2. The median fold expression of IFN-β (TB- 0.030; HHC- 0.031) and IFN-γ (TB- 0.71; HHC- 0.79) were similar in TB and HHC. However, IFN-α transcripts were ~3.5 folds higher ($p<0.0001$) in TB patients (0.8 median fold,) compared to HHC (0.23 medial fold; Fig 1; Table 2). High expression of IFN-α was seen among 46% (56/ 123) of the TB patients. Similar levels of mRNA expression for IFN-α was limited to a lower percentage (26%, 22/86) of HHC.

### Effective ATT alters the expression of interferons in patients treated

The expression levels of IFN-α, IFN-β, and IFN-γ in 36 TB patients at the time of enrolment and following successful completion of DOTS treatment were compared (Figs 2 & 3). A

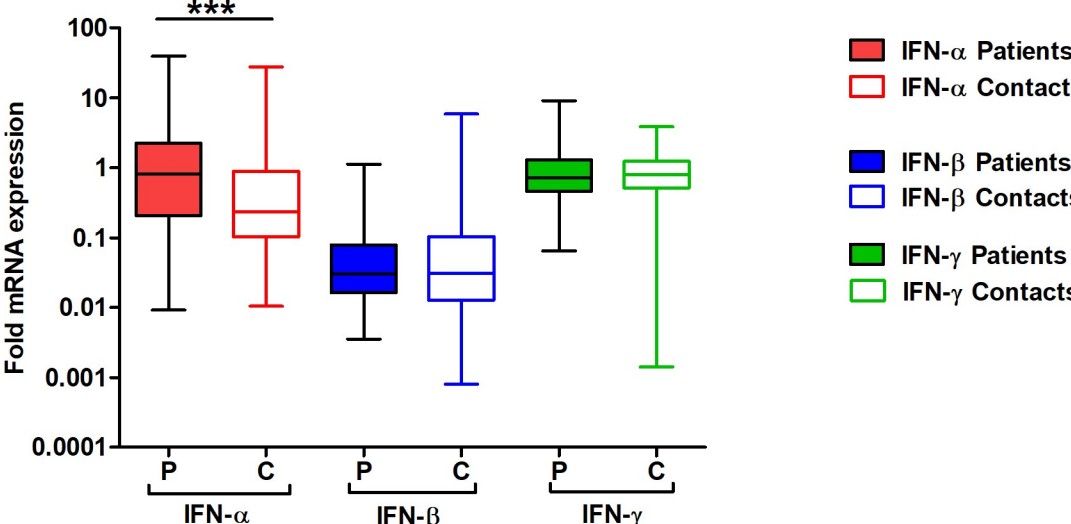

**Fig 1. Box plot shows the fold mRNA expression of IFN-α, IFN-β and IFN-γ in untreated TB patients (n = 123) compared to healthy family contacts of patients (n = 86) as estimated by Real-Time PCR.** Target gene expression was normalized with β-actin gene expression. The data has been calculated with the $2^{-\Delta\Delta Ct}$ formula, as described in methods. The fold mRNA expression has been determined with reference to healthy laboratory volunteers. The horizontal bar represents the median value for mRNA in each group, the 25th and 75th percentile have been represented by the boxes. ⊥ & ⊤—the whiskers represent the maximum and minimum values of the data, respectively. The data has been plotted in log₁₀ scale. To compare the transcripts level between groups, non-parametric Mann Whitney test was applied, ***$p < 0.0001$, IFN-α expression in TB patients Vs Healthy family contacts.

**Table 2. Comparative analysis of fold mRNA expression detected for interferons among TB patients and healthy family contacts.**

| Group | Fold mRNA expression of Interferon@ | | | | | | | | |
|---|---|---|---|---|---|---|---|---|---|
| | IFN-α | | | IFN-β | | | IFN-γ | | |
| | Low | Median | High | Low | Median | High | Low | Median | High |
| TB Patients (n = 123) | 0.2066 | **0.8036***  | 2.2420 | 0.01626 | 0.03035 | 0.07899 | 0.4588 | 0.7101 | 1.2980 |
| Healthy family Contacts (n = 86) | 0.1022 | **0.2348** | 0.8931 | 0.01270 | 0.03099 | 0.1032 | 0.5145 | 0.7934 | 1.230 |

(@- fold mRNA expression calculated with reference to healthy laboratory volunteers

*—$p < 0.0001$ IFN-α expression in TB patients Vs Healthy family contacts)

substantial decrease in the expression levels of IFN-α and IFN-β (Fig 2A & 2B) was observed. IFN-α expression levels were ~3-fold reduced in treated patients (0.13 median fold) compared to untreated patients (0.37; Fig 2A; $p< 0.001$). Similarly, IFN-β levels were also ~2-fold lower in treated patients (0.04) compared to untreated patients (0.02; Fig 2B; $p < 0.001$). However, IFN-γ expression was sustained and unaltered in patients as assessed at the time of enrolment and on successful completion of treatment (Fig 2C).

## Modulation of IFN-α levels in TB patients by PGE2 and Interferon Regulatory Factors (IRFs)

We evaluated some critical regulators of IFN-α expression, namely PGE2 and IRF5 & 7 among TB patients and HHCs. Prostaglandin E2 (PGE2) is an eicosanoid derivative, which functions as a potent inhibitor of Type I IFNs [21]. As expected, we found in individuals expressing (>1 fold) high levels of IFN-α (in both TB patients and HHC groups), had diminished serum levels of PGE2 and vice versa (Fig 4A & 4B). This inverse relationship between levels of PGE2 and IFN-α was significant (Spearman's rho = -0.563; $p<0.0001$, Fig 4C).

Amongst the Interferon Regulatory Factors (IRFs) that transcribe Type I IFNs, IRF7 is critical [22]. Although IRF5 promotes the IFN-γ pathway through direct induction of IL-12 [23,24], reports indicate that it may be necessary for the transcription of Type I IFNs as well [25,26]. Levels of IRF5 amongst the high (n = 30, TB patients) and low (n = 32, TB patients) IFN-α expressers, were similar (Fig 5), indicating IRF5 may not be directly involved in Type I IFN modulation. In agreement with published reports [27–29], the expression of IRF7 was significantly elevated (~967 folds higher; median fold expression $60 \times 10^{-4}$) in high IFN-α expressing patients, compared to low IFN-α expressing patients, ($0.062 \times 10^{-4}$; $p > 0.0001$; Fig 5).

## Temporal expression of IFN-α, IFN-β, and IFN-γ in THP1 cells infected with live or heat-killed M.tb

Our results demonstrated that successful treatment of TB patients results in a significant reduction of IFN-α and IFN- β expression (Figs 2 and 3). The impact of these changes on the survival of intracellular *M.tb* in human macrophages *in vitro* was assessed. Elevated levels of IFN-α compared to IFN-γ and β in live *M.tb* infected THP-1 cells was observed at 24 hours (S2 Fig, Panel A). The conditions seen among treated and untreated TB patients were mimicked *in-vitro* by the addition of specific antibody into designated wells containing infected cells. The specific antibody neutralized the biological activity of each IFN present in the milieu of infected THP-1 cells. The enumeration of CFUs confirmed the presence of viable intracellular *M.tb* in the treated cell cultures.

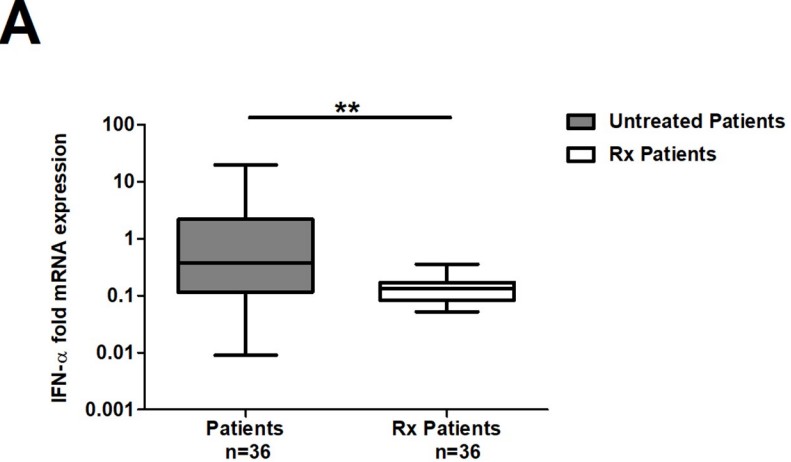

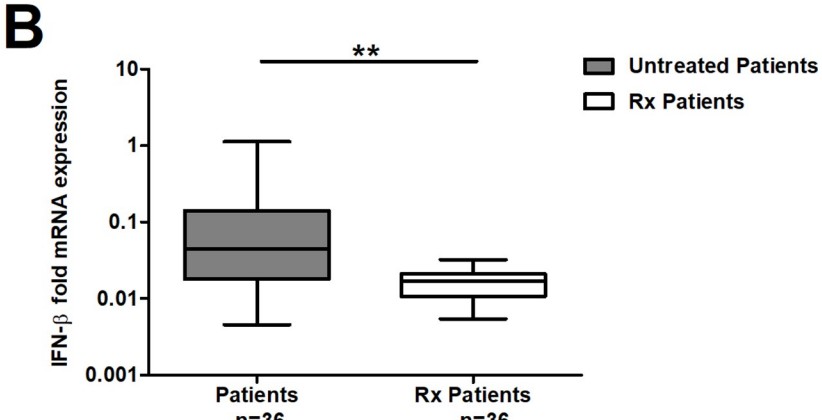

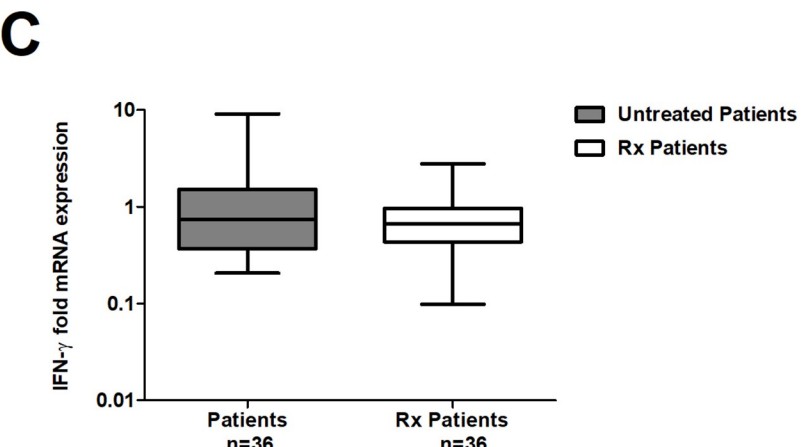

**Fig 2. Box plot depicting comparative analysis of mRNA expression of IFNs detected in paired samples obtained at the time of recruitment and after successful completion of treatment from 36 patients.** Panel A, IFN-α; Panel B, IFN-β; and Panel C, IFN-γ. Target gene expression was normalized with β-actin gene expression. The data has been calculated with the $2^{-\Delta\Delta Ct}$ formula, as described in methods. The horizontal bars represent the group median, longitudinal bars represent minimum and maximum values (⊥ & ⊤). The data has been plotted in $\log_{10}$ scale. To

compare the transcript levels between groups, non-parametric Mann Whitney test was applied, **- $p < 0.0001$, IFN-α expression baseline Vs After treatment; *- $p < 0.001$, IFN-β expression baseline Vs After treatment.

## IFN-α enhances mycobacterial growth

Anti-IFN-α neutralizing antibody significantly reduced the CFUs (5,595 ± 412.22 CFUs) compared to untreated infected control cells (24,166.33 ± 1687.68 CFUs, $p < 0.0001$; Fig 6). In contrast, infected cultures treated with anti-IFN-γ antibody, the *M.tb* CFUs were significantly higher (39,880.67 ± 5806.31CFUs) than the control cultures, (24,166.33 ±1687.68 CFUS, $p < 0.001$, Fig 6). Whereas in infected cultures treated with either anti-IFN-β antibody alone (21,428 ±5893.4 CFUs) or simultaneously with a mixture anti-IFN-α, anti-IFN-β and anti-IFN-γ neutralizing antibodies, the enumerated *M.tb* CFUs (22,142±3092.87 CFUs) did not differ significantly from the control cultures, (24,166.33 ± 1687.68 CFUs). The lowest number of viable *M.tb* was obtained in cultures treated with anti-IFN-α antibody (6071±412.2 CFUs) compared to all other cultures exposed individually or in combination to the panel of neutralizing antibodies. The highest number of viable *M.tb* was obtained in cultures treated with anti-IFN-γ antibody (46,428±5806.3 CFUs).

## Discussion

Results of this study showed that of the three IFNs examined, IFN-α mRNA expression levels predominated compared to IFN-γ and IFN-β among untreated TB patients. The significant maximal median fold of mRNA IFN-α expression is among untreated TB patients compared to family contacts clinically free of tuberculosis. However, the levels of IFN-α expression among the TB patients varied. On the other hand, following successful treatment, expression levels of IFN-α, and IFN- β regressed, resulting in the predominance of IFN-γ. The circulatory levels of PGE2 and IRF expression influenced the expression levels of IFN-α. These observations showed that IFN-α expression was indeed associated with ongoing active clinical tuberculosis, and the levels reduced or absent in patients following clearance and reduction in mycobacterial load induced by effective chemotherapy. Further, using *in-vitro M.tb* infected THP-1 cells, it was observed neutralization of IFN-α enabled an effective reduction in mycobacterial viability. In contrast, the presence of IFN-α / β in the absence of IFN-γ leads to enhanced viability of intracellular *M. tb*.

Production of type I IFNs in cultured peripheral blood monocytes of patients with active TB and the inducible transcriptional signature of type I IFNs in blood leukocytes derived from active TB patients has been reported [30,31]. Among untreated TB patients, a higher number of pDCs in circulation, the principal source of IFN-α is reported [32]. In mice, copious quantities of Type I IFN is induced by virulent strains compared to non-virulent strains of *M.tb* [33,34]. Mycobacterial constituents and extracellular mycobacterial DNA (eDNA) bound to Toll-like and cytosolic receptors, respectively, induce IFN-α [33–37]. Furthermore, the induction of IFN-β varies with the infecting strain of *M.tb* ability to induce the release of host mitochondrial DNA [38]. The concentration of PGE2 and positively regulating transcriptional factors, namely IRF5 & IRF7, modulate Type I IFN expression. An inverse relationship exists between the amount of PGE2 in circulation and the levels of IFN-α mRNA; high IFN-α expressers with low levels of PGE2 and vice versa. Elevated levels of Type I IFN limit and inhibit PGE2 production both *in vitro* as well as *in vivo* [39]. IRF7 expression was limited to high IFN-α expressers, whereas IRF5 expression was minimal. These differences in the expression of transcriptional factors influenced by the sensing mechanism, and the availability of the appropriate ligand, bias the expression level(s) of the transcriptional factors and the IFN-α

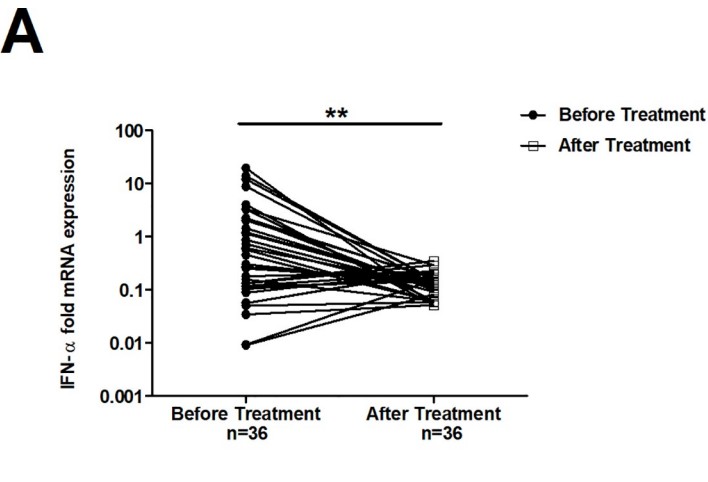

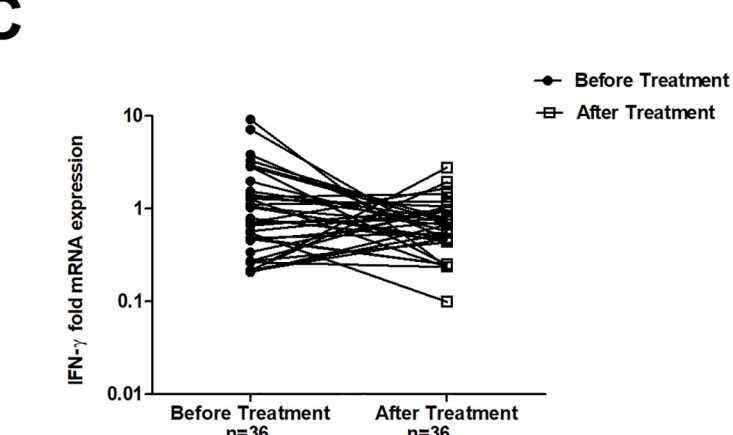

**Fig 3. Line graph depicting mRNA expression of IFNs detected in individual paired samples at the time of recruitment and after successful completion of treatment, (N = 36 patients).** Panel A: IFN-α; Panel B: IFN-β; and Panel C: IFN-γ. Target gene expression was normalized with β-actin gene expression. The fold mRNA expression has been calculated with the $2^{-\Delta\Delta Ct}$ formula, as described in methods. The data has been plotted in $\log_{10}$ scale. ** $p < 0.001$- IFN-α expression before and after treatment, *** $p < 0.0001$- IFN-β expression before and after treatment.

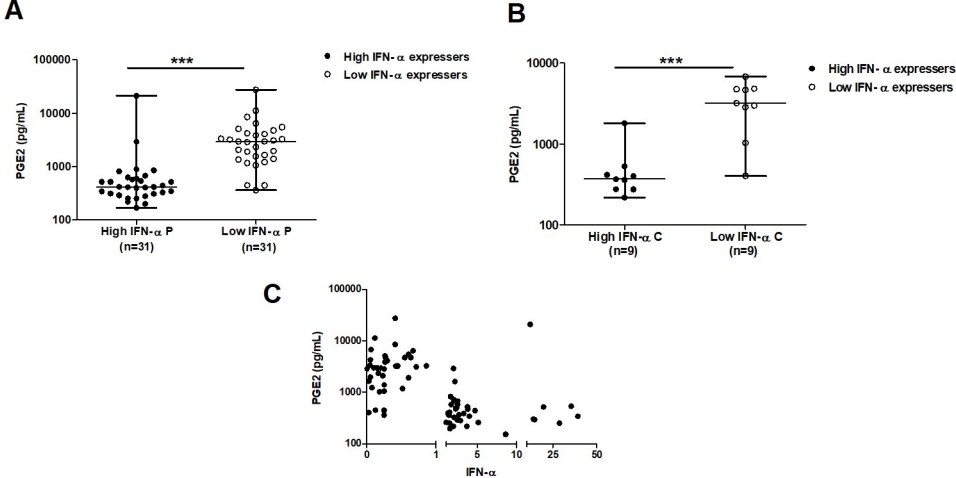

**Fig 4.** Scatter plot shows the inverse relationship between IFN-α expression and circulating levels of Prostaglandin E2 among TB patients (Panel A) and household family contacts (Panel B). High IFN-α expression: fold mRNA expression >1; low IFN-α expression: fold mRNA expression <1. The data has been plotted in $log_{10}$ scale. Non-parametric Mann Whitney test was applied, ***- $p$<0.0001, P—TB patients expressing IFN-α High Vs Low expressers; **-$p$<0.001, C-Family Contacts expressing IFN-α High Vs Low expressers. **Panel C:** Correlation plot between IFN-α expression and circulating levels of Prostaglandin E2 among TB patients. Non-parametric Spearman's rho = -0.563; $p$<0.0001.

subtype generated [40]. The cytosolic sensor, namely the nucleotide-binding oligomerization domain (NOD) 2 mediates expression of IRF5. The ligand associated with NOD2 induced production of Type I IFN is muramyl dipeptide (MDP) [41,42], whereas the ligand for stimulating expression of IRF7 is nucleic acids sensed by cytosolic receptors [40].

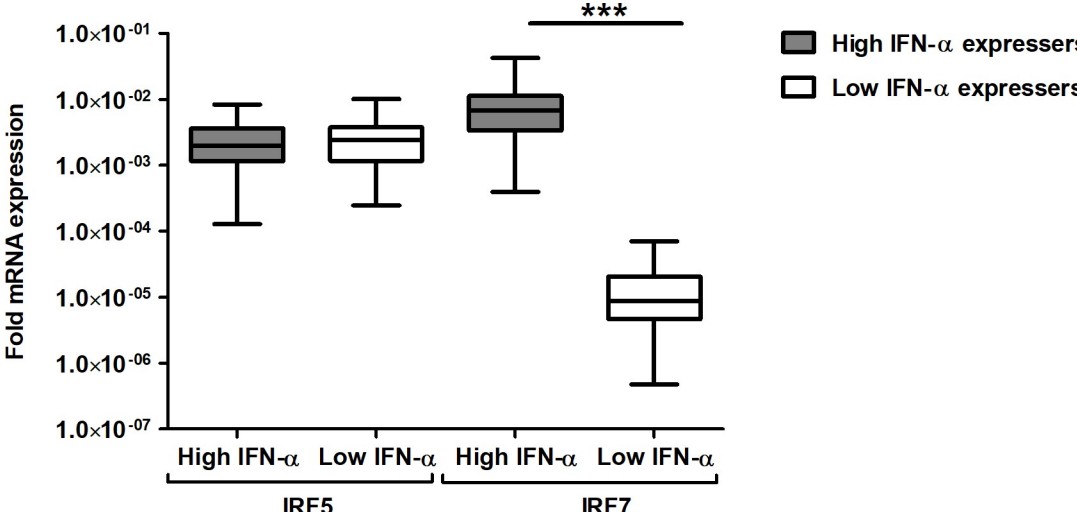

**Fig 5. Boxplot shows the fold mRNA expression of transcriptional factors IRF5 and IRF7 among TB patients.** Comparative fold mRNA expression of transcriptional factors IRF5 and IRF7 among TB patients segregated into High (fold mRNA expression >1) and low (fold mRNA expression <1) IFN-α expressers. Target gene expression was normalized with β-actin gene expression. The data has been calculated with the $2^{-\Delta\Delta Ct}$ formula, as described in methods. The fold mRNA expression has been determined with reference to healthy volunteers. The horizontal bar represents the median value for mRNA in each group, the 25th and 75th percentile have been represented by the boxes. ⊥ & ⊤—the whiskers represent the maximum and minimum values of the data, respectively. The data has been plotted in $log_{10}$ scale. To compare the transcripts level between groups, non-parametric Mann Whitney test was applied. ***- $p < 0.0001$, IRF 7 expression among high IFN-α expressers Vs Low IFN-α expressers.

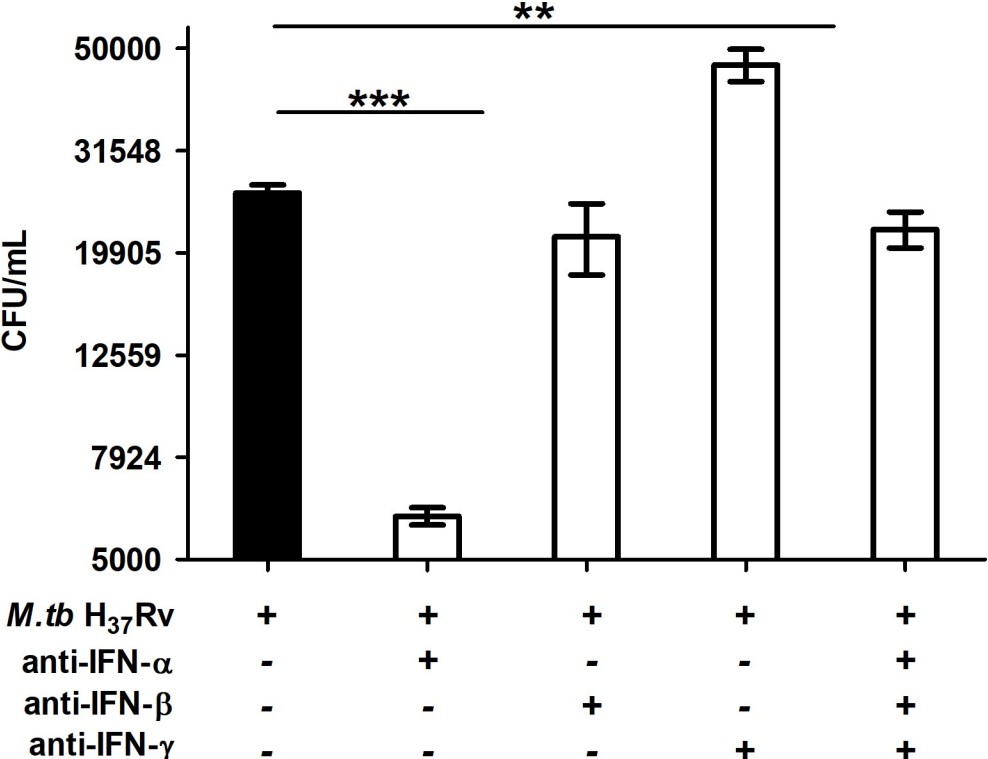

**Fig 6. Histogram depicting the effect of blocking IFNs with specific neutralizing antibodies on survival of intra cellular _M.tb_ present in THP-1 cells, as determined by colony forming units (CFU).** Differentiated THP-1 cells were infected with _M.tb_ (10:1 MOI). The extracellular bacteria were removed and the respective neutralizing anti-IFN antibodies were added as described in methods. The cells were harvested on the 5th day and processed for CFU estimation. The bars represent the mean CFU ±SD of three independent experiments. The ***—$p<0.0001$, CFU of Control Vs CFU of _M.tb_ infected cells $R_x$ with anti-IFN-α antibodies, **—$p<0.001$, CFU of Control Vs CFU of _M.tb_ infected cells $R_X$ with anti-IFN-γ antibodies. Students t Test.

IFNs influence viability of intracellular _M.tb_. In the current study, the i*n-vitro* neutralization of IFN-α enables efficient mycobactericidal activity facilitating a reduction in mycobacterial viability, despite the presence of IFN- β. In contrast, the presence of IFN-α/ β in the absence of IFN-γ leads to enhanced viability of intracellular _M.tb_, as seen by the highest number of CFUs recovered from these cultures. These results showed that alterations in the bioactivity of IFNs influence intracellular survival of _M.tb_, as evidenced by the heightened efficacy in the killing of _M.tb_ in the absence of IFN-α and enhanced viability of _M.tb_ seen in the absence of IFN-γ. The presence or neutralization of IFN-β did not influence the survival/killing of intracellular _M.tb_. These deliberate _in vitro_ alterations emphasize the significance of high levels of IFN-α expressed by patients before treatment (during active disease, Fig 1, Table 2) potentially impact IFN-γ mediated anti-mycobacterial activity adversely during active disease; compared to the repression of IFN-α expression leading to the predominance of IFN-γ that occurs after successful treatment. Several reports suggest IFN-α exacerbates _M.tb_ infection in multiple ways [31,33,43]. IFN-α mediates the downregulation of potentially protective cytokines such as TNF-α, IL-1β & IL-12 and promotes the secretion of IL-10 an immune-suppressive cytokine [44]. IFNAR-/- mice have enhanced and late mortality compared with WT mice [34]. IFN-α suppresses IFN-γ mediated killing of mycobacteria both by an IL-10 dependent [45], as well as in an independent manner. IFN-α disrupts IFN-γ mediated activation of macrophages, compromising anti-mycobacterial activity, by downregulating IFN-γ receptor

expression [46]; down-regulation of IFN-γ receptor expression following *M.tb* infection has been reported [47]. *In-vitro M.tb* THP-1 model of infection showed the predominance of IFN-α, similar to that seen among untreated TB patients. The predominance of IFN-α would enhance mycobacterial viability leading to *in-vivo* overburdening and chronicity of infection. In contrast, the dominance of IFN-γ, as seen in treated TB patients, would lead to the augmented killing of intracellular *M.tb*, and *in-vivo* clearance, [48,49].

The current study showed that IFN-α expression was indeed associated with ongoing clinical tuberculosis and is reduced or absent following successful treatment. These observations suggest that the assessment of Type I IFNs levels would help monitor the patient's response to chemotherapy. Hence assessment of Type I IFNs would be advantageous compared to the detection of acid-fast bacilli (AFB) by conventional smear microscopy, as changes in the expression of Type I IFNs are expected to occur earlier and to precede the clearance and reduction in the mycobacterial load. Early and reliable assessment of response to chemotherapy would benefit the clinical management of tuberculosis patients.

## Supporting information

**S1 Fig. Histogram depicting expression of interferon and interferon inducible genes in the presence of specific neutralizing anti-interferon antibodies.** (**Panel A**) *IFI44* (type I interferon inducible gene), and (**Panel B**) *FcγR1* (type II interferon inducible gene) was monitored in PMA differentiated THP-1 cells. The gene expression was normalized with β-actin gene expression. The fold expression was calculated as described in methods. The bars represent the mean fold mRNA expression ± SD of three independent experiments. The data has been plotted in $\log_{10}$ scale. *** —$p < 0.0001$, *IFI44* and *FcγR1* expression in control Vs cells treated with neutralizing anti-IFN-α and anti-IFN-γ antibodies respectively, Students t Test.
(TIF)

**S2 Fig.** Time kinetics of mRNA expression as assessed by real-time PCR of type I and type II interferons in PMA-differentiated THP1 cells infected with *M.tb* (Panel A: Live; Panel B: Heat-killed). RNA was extracted at indicated time points and subjected to real-time PCR. Target gene expression was normalized with β-actin gene expression. The data has been calculated with the $2^{-\Delta\Delta Ct}$ formula, as described in methods and has been plotted in $\log_{10}$ scale.
I-Mean ± SD.
(TIF)

**S1 Data.**
(XLSX)

## Acknowledgments

The fellowship to Ms Vibha Taneja from Indian Council of Medical Research (ICMR), Government of India; H.K.P Emeritus Medical Scientist, ICMR,; Financial support of the ICMR grant "Study of Immunological impact of type I Interferon (s) on immune response in tuberculosis patients," project ID: 2013–0063; Drs. Jaya Sivaswami Tyagi, Department of Biotechnology, Ashish Datt Upadhyay and V. Sreenivas, Department of Biostatistics, All India Institute of Medical Sciences, New Delhi; Bio-informatics facility of the Biotechnology Department and the technical assistance of Mr. Krishan Pal Singh, and Shailendra Kumar. *M.tb* $H_{37}Rv$ was a gift from Dr. Richard F. Silver, Case Western Reserve University, Cleveland, OH, USA. The THP-1 cell line was a gift from Dr. Nasreen J. 353 Ehtesham, National Institute of Pathology, New Delhi.

## Author Contributions

**Conceptualization:** Hanumanthappa Krishna Prasad.

**Formal analysis:** Hanumanthappa Krishna Prasad.

**Funding acquisition:** Hanumanthappa Krishna Prasad.

**Investigation:** Vibha Taneja, Manish Goel, Gopi Chand Khilnani, G. B. K. S. Prasad, Hanumanthappa Krishna Prasad.

**Methodology:** Priya Kalra, Gopi Chand Khilnani, Vikram Saini, Hanumanthappa Krishna Prasad.

**Project administration:** Gopi Chand Khilnani, Hanumanthappa Krishna Prasad.

**Resources:** Gopi Chand Khilnani, Umesh Datta Gupta, Hanumanthappa Krishna Prasad.

**Supervision:** Gopi Chand Khilnani, Hanumanthappa Krishna Prasad.

**Validation:** Hanumanthappa Krishna Prasad.

**Visualization:** Vibha Taneja.

**Writing – original draft:** Vikram Saini, Hanumanthappa Krishna Prasad.

**Writing – review & editing:** Priya Kalra, Hanumanthappa Krishna Prasad.

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
