## [Decision Letter · Decision Letter 0]

4 Mar 2020

PONE-D-20-02309

Impact and Prognosis of the Expression of IFN-α Among Tuberculosis Patients

PLOS ONE

Dear Dr. Hanumanthappa,

Thank you for submitting your manuscript to PLOS ONE. After careful consideration, we feel that it has merit but does not fully meet PLOS ONE’s publication criteria as it currently stands. Therefore, we invite you to submit a revised version of the manuscript that addresses the points raised during the review process.

Please carefully address all  comments/ suggestions from the reviewer. 

We would appreciate receiving your revised manuscript by Apr 18 2020 11:59PM. To enhance the reproducibility of your results, we recommend that if applicable you deposit your laboratory protocols in protocols.io, where a protocol can be assigned its own identifier (DOI) such that it can be cited independently in the future. For instructions see: http://journals.plos.org/plosone/s/submission-guidelines#loc-laboratory-protocols

We look forward to receiving your revised manuscript.

Kind regards,

Martin E Rottenberg

Academic Editor

PLOS ONE

Journal Requirements:

2. Please provide additional details regarding participant consent. In the ethics statement in the Methods and online submission information, please ensure that you have specified whether consent was written or verbal/oral. If consent was verbal/oral, please specify: 1) whether the ethics committee approved the verbal/oral consent procedure, 2) why written consent could not be obtained, and 3) how verbal/oral consent was recorded. If your study included minors, please state whether you obtained consent from parents or guardians in these cases.

Reviewers' comments:

Reviewer's Responses to Questions

**Comments to the Author**

1. Is the manuscript technically sound, and do the data support the conclusions?

Reviewer #1: Partly

2. Has the statistical analysis been performed appropriately and rigorously? 

Reviewer #1: Yes

3. Have the authors made all data underlying the findings in their manuscript fully available?

Reviewer #1: No

4. Is the manuscript presented in an intelligible fashion and written in standard English?

Reviewer #1: No

5. Review Comments to the Author

Reviewer #1: Taneja et al describe a study of interferon gene expression in Tuberculosis patients. The study is of interest and relevant as many questions still remain related to the role of type I interferons in TB. My main concern relates to the main findings presented and how the data is analyzed and interpreted.

The authors measure different IFNs by quantifying gene expression by RT-PCR and normalizing to B-actin and expressing relative values as 2-ddCT. High expression was defined as mRNA expression of IFN >1 fold, however the results plotted show median values of less than 1 (Figure 1 & 2). There may be difference between the patient and controls for these values, but what does that mean if the fold expression is less than 1 ?

In parallel from the in vitro challenge model the fold expression is indeed much greater than 1, between 100-1000 fold. This makes the ex vivo data very challenging to interpret.

Related how was the house-keeping gene selected ? Where there any differences in its expression between the different groups examined ? Have the authors tested whether these modest gene expression differences reflect differences in circulating immune cell proportions?

Also how do the authors reconcile their findings with the literature where although ISG signatures have been identified in TB disease by many groups, elevated IFN gene expression (Berry et al, Nature 2010) or more recently elevated protein levels (Llibre et al, Front Cell Infect Microbiol. 2019) have not been reported.

Minor

- How were patients defined to have TB disease ?

- Which IFNa subtype is detected with this assay ?

- Figures 1 & 2 should be plotted with a log scale as the break in the axis is confusing.

- Fig 4 should be plotted on log scale as many points are too close to the threshold, and the correlation plot should be included

- How do the authors explain that blocked IFNa alone inhibits bacterial growth, but not when given with anti-IFNB and anti-IFNg ? Do THP1 cells express IFNGR and IFNAR ? To equal levels ?

6. PLOS authors have the option to publish the peer review history of their article (what does this mean?). If published, this will include your full peer review and any attached files.

Reviewer #1: No

---

## [Author Response · Author response to Decision Letter 0]

15 May 2020

The comments and suggestions made by the Reviewer has been helpful and their inclusion has vastly improved the manuscript presentation etc.. My point by point response to the suggestions and comments are detailed in the file 'Response to Reviewers.'

---

## [Editor Report · Decision Letter 1]

17 Jun 2020

Impact and Prognosis of the Expression of IFN-α Among Tuberculosis Patients

PONE-D-20-02309R1

Dear Dr. Hanumanthappa,

We’re pleased to inform you that your manuscript has been judged scientifically suitable for publication and will be formally accepted for publication once it meets all outstanding technical requirements.

Kind regards,

Martin E Rottenberg

Academic Editor

PLOS ONE
---

## [Editor Report · Acceptance letter]

26 Jun 2020

PONE-D-20-02309R1 

Impact and Prognosis of the Expression of IFN-α Among Tuberculosis Patients 

Dear Dr. Krishna Prasad:

I'm pleased to inform you that your manuscript has been deemed suitable for publication in PLOS ONE. Congratulations! Your manuscript is now with our production department. 

Kind regards, 

on behalf of

Dr. Martin E Rottenberg 

Academic Editor

PLOS ONE